# How prior preferences determine decision-making frames and biases in the human brain

Alizée Lopez-Persem[1,2], Philippe Domenech[2,3,4], Mathias Pessiglione[1,2]*

[1]Motivation, Brain and Behavior lab, Centre de NeuroImagerie de Recherche, Institut du Cerveau et de la Moelle épinière, Paris, France; [2]Inserm U1127, CNRS U 7225, Université Pierre et Marie Curie, Paris, France; [3]CHU Henri Mondor, DHU Pe-PSY, Service de Neurochirurgie Fonctionnelle, Créteil, France; [4]Behavior, Emotion and Basal Ganglia lab, Institut du Cerveau et de la Moelle épinière, Paris, France

**Abstract** Understanding how option values are compared when making a choice is a key objective for decision neuroscience. In natural situations, agents may have a priori on their preferences that create default policies and shape the neural comparison process. We asked participants to make choices between items belonging to different categories (e.g., jazz vs. rock music). Behavioral data confirmed that the items taken from the preferred category were chosen more often and more rapidly, which qualified them as default options. FMRI data showed that baseline activity in classical brain valuation regions, such as the ventromedial Prefrontal Cortex (vmPFC), reflected the strength of prior preferences. In addition, evoked activity in the same regions scaled with the default option value, irrespective of the eventual choice. We therefore suggest that in the brain valuation system, choices are framed as comparisons between default and alternative options, which might save some resource but induce a decision bias.

*For correspondence: mathias. pessiglione@gmail.com

**Competing interests:** The authors declare that no competing interests exist.

## Introduction

Standard decision theory assumes that when faced with a choice, individuals first assign subjective values to each option, and then compare these values in order to select the best option (*Samuelson, 1938*; *Von Neumann and Morgenstern, 1947*). Understanding the neural mechanisms governing this valuation/selection process has become a central aim in the field of decision neuroscience. A large set of fMRI evidence points to the ventro-medial Prefrontal Cortex (vmPFC) as a key player in the valuation process (*Bartra et al., 2013*; *Clithero and Rangel, 2014*). Neural activity in the vmPFC reflects subjective values, either measured with likeability ratings or inferred from binary choices (*Kable and Glimcher, 2009*; *Rangel and Hare, 2010*). In accordance with the idea of a common neural currency (*Levy and Glimcher, 2012*), the vmPFC was found to encode the subjective value of many kinds of goods, such as food, money, trinkets, faces, paintings, charities, etc. (*Chib et al., 2009*; *Hare et al., 2010*; *Lebreton et al., 2009*; *Plassmann et al., 2007*). Such value coding was observed not only during choice but also in the absence of choice, during passive viewing of items presented in the attentional focus or when performing a distractive task on these items (*Lebreton et al., 2009*; *Levy et al., 2011*; *Abitbol et al., 2015*).

During binary choices, it has been repeatedly shown that vmPFC activity correlates with the relative value of the two options under consideration ($V_A$–$V_B$). However, the framing of such decision value signal, i.e. what A and B actually represent, remains an unresolved issue. This question is of importance because the brain regions downstream in the decision process cannot operate the appropriate selection without knowing which option is favored by the relative value signal. In

**eLife digest** If you had the choice of listening to a piece of music by either the singer Céline Dion or jazz pianist Keith Jarrett, which would you pick? When choosing between two mutually exclusive options, the brain first assigns a value to each. An area called the ventromedial prefrontal cortex (vmPFC) compares these two values and calculates the difference between them. The vmPFC then relays this difference to other brain regions that trigger the movements required to obtain the selected option.

But what exactly is the vmPFC comparing? A reasonable assumption is that we approach the decision with an existing preference for one of the options based on our previous experience. Lopez-Persem et al. set out to determine whether and how the vmPFC uses this existing preference – for example, for pop music over jazz – to drive the decision-making process.

For the experiments, volunteers were asked to rate how much they liked individual musicians spanning a range of different genres. While lying inside a brain scanner, the subjects then had to choose their favorite from pairs of musicians selected from the list. When making such decisions, volunteers must consider both the overall category (do I prefer jazz or pop?) but also the individual examples (a pop music fan might choose jazz if the pop option is Britney Spears). Lopez-Persem et al. found that the volunteer's decisions were biased towards their prior preference. Pop music fans chose Céline Dion or Britney Spears more often than would be expected based on the likability ratings they had given the individual artists in the study.

Brain imaging revealed that the vmPFC represents choices as 'default minus alternative', where the default is any member of the previously preferred category (e.g. any pop artist for a pop music fan) and the alternative is from a different category (e.g. a jazz artist). Baseline vmPFC activity is higher for members of the preferred category, giving these options a head start over the alternatives. Asking volunteers to choose between other types of objects, including food and magazines, produced similar results. The brain thus uses a general strategy for decision-making that saves time and effort, but which also introduces bias. The next step is to work out how downstream brain regions use the vmPFC signal to select the preferred option.

particular, the post-decisional frame that has often been reported (*Boorman et al., 2013*; *Hare et al., 2011*; *Hunt et al., 2012*) provides a decision value signal between chosen and unchosen options ($V_{ch}-V_{unch}$) that cannot be used for making the selection. A spatial frame, based on the location of options (e.g., $V_{left}-V_{right}$), has been suggested but not supported by much experimental evidence regarding the vmPFC valuation signal (*Palminteri et al., 2009*; *Wunderlich et al., 2009*; *Skvortsova et al., 2014*). A more promising suggestion is the attentional frame (*Krajbich et al., 2010*), in which the decision value signal encoded in the vmPFC depends on which option is attended to ($V_{att}-V_{unatt}$). Such framing provided a good account for vmPFC activity in a choice task where fixation patterns were imposed to subjects, and correctly predicted several features of spontaneous choice behavior by imposing a discount weight on the unattended option value (*Krajbich et al., 2010*; *Lim et al., 2011*). Notably, the attentional frame predicts that more fixated options should be more frequently chosen, which might explain why vmPFC activity has been found to correlate with $V_{ch}-V_{unch}$ in other studies. However, the attentional model assumes that visual exploration is random, which might be true in artificial laboratory tasks where subjects have no information about upcoming options, but not in natural situations where prior knowledge might play a role.

Here, we hypothesize that the framing of the decision value encoded in the vmPFC is imposed by prior preferences. In other words, vmPFC activity should scale positively with the value of the option that is preferred a priori, which we call the default option, and negatively with that of the alternative ($V_{def}-V_{alt}$). This hypothesis is compatible with the observation that vmPFC activity correlates with $V_{ch}-V_{unch}$, since choices usually follows on prior preferences. Yet, the interpretation is fundamentally different, as $V_{def}-V_{alt}$ is a pre-decisional value signal susceptible to drive option selection. Our hypothesis builds on the literature about optimal foraging, which argues that stay/switch choice is the natural case of decision-making (*Stephens and Krebs, 1986*). In this

framework, staying on a same patch is the default option against which all alternatives must be compared. Several studies investigated such stay/switch decisions and implicated the dorsal anterior cingulate cortex in promoting a shift away from the default option (*Hayden et al., 2011*; *Kolling et al., 2012*; *Kvitsiani et al., 2013*), while others induced default policies by manipulating prior probabilities of being correct (*Boorman et al., 2013*; *Fleming et al., 2010*; *Mulder et al., 2012*; *Scheibe et al., 2010*). Although experimental manipulations vary across these studies, the default option is always defined as the option that would be selected in the absence of further information processing about its value relative to the alternatives. This definition provides objective criteria to identify the default option in a choice set: it should be selected faster and more frequently than the alternatives.

Therefore, our hypothesis implies that prior preferences should (1) induce a bias in favor of the default option, and (2) determine the frame of the value comparison process. The purpose of the present study was to examine how these two constraints would shape the brain valuation signal. To do so, we exploited the hierarchical structure of preferences: individuals have global preferences between categories of goods that can be locally reversed when comparing particular items. For instance, someone may prefer pop to jazz music in general, but nonetheless pick Keith Jarrett if the only other option is Britney Spears. In a binary choice, the prior preference at the category level thus designates a default option (i.e., the item belonging to the preferred category), but the option values still need to be compared at the item level in order to reach a final decision.

We conducted an fMRI experiment where participants made binary choices between items belonging to different categories. Preferences between categories were inferred from likeability ratings that were collected for every item before the scanning session. In the following analyses, we first establish the presence of a bias toward the default option in both choice and response time, above and beyond the prior preference between categories. Using computational modeling, we provide evidence that the default bias is best accounted for by a shift in the starting point of a drift diffusion process, which is proportional to the prior preference between categories. Then, we show that the default bias is unrelated to gaze fixation pattern, precluding an attentional framing. Finally, we uncover two effects of prior preference in fMRI data: (1) vmPFC baseline activity reflects the a priori shift in favor of the default option, and (2) vmPFC evoked response represents the value of the default option, irrespective of the eventual choice.

## Results

### Behavior

Prior to the scanning session, participants (n = 24) rated the likeability of items belonging to three different domains (food, music, magazines). Each domain included four categories of 36 items (see Materials and methods). At that time, participants were unaware of these categories. This is because the presentation of items for likeability ratings was blocked by domain but not by categories, which were randomly intermixed. During the scanning session, subjects performed series of choices between two items (*Figure 1*), knowing that one choice in each domain would be randomly selected at the end of the experiment and that they would stay in the lab for another 15 min to enjoy their reward (listening to the selected CD, eating the selected food and reading the selected magazine). Trials were blocked in a series of nine choices between items belonging to the same two categories within a same domain. The two categories were announced at the beginning of the block, such that subjects could form a prior preference (although they were not explicitly asked to do so). We quantified this prior preference as the difference between mean likeability ratings (across all items within each of the two categories), which is hereafter denoted as $DV_{CAT}$. In most cases (84 ± 3% on average), preferences inferred from mean ratings matched the preferences between categories that subjects directly expressed in post-scanning debriefing tasks. Moreover, the confidence in these choices between categories, which subjects provided on an analog rating scale during debriefing, was significantly correlated to $DV_{CAT}$ (r = 0.44 ± 0.06, t(23) = 7.88, p = $5.10^{-8}$). These explicit measures taken after the scanning session therefore validate our quantification of implicit preferences between categories. In the following, we analyze choices and response times to assess the presence of a bias in favor of the default option (i.e., the item belonging to the preferred category).

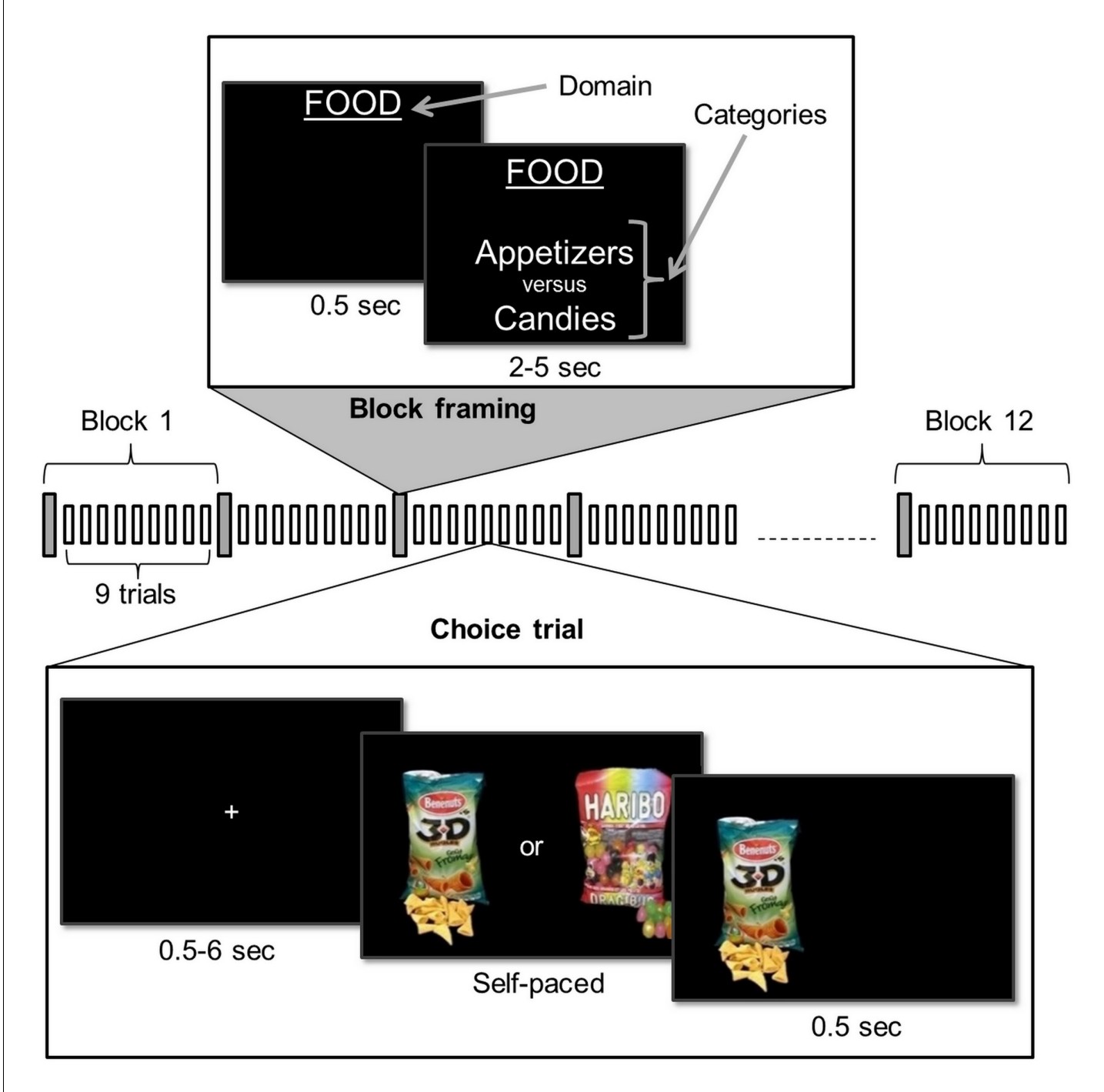

**Figure 1.** Choice task. Participants performed the choice task inside the MRI scanner. The task was composed of four 12-block sessions. During a block, subjects first saw an instruction screen indicating the reward domain (e.g., food) and the two categories from which choice options were drawn. Then, they had to make a series of nine binary choices, each confronting the two categories with two novel items. The choice was self-paced and feedback on chosen option was provided to the subject.

We fitted a simple logistic regression model including a constant, the default option value, denoted $V_{IT}(def)$, and the alternative option value, denoted $V_{IT}(alt)$, to choices expressed in the 'default vs. alternative' frame. Regression coefficient estimates showed that the two option values were equally contributive to the choice ($V_{IT}(def)$: $\beta = 0.060 \pm 0.005$, $t(23) = 11.90$, $p = 3.10^{-11}$;

$V_{IT}(alt)$: $\beta = -0.060 \pm 0.004$, $t(23) = -14.21$, $p = 7.10^{-13}$). Crucially, the constant was significantly positive ($\beta = 0.68 \pm 0.13$, $t(23) = 5.40$, $p = 2.10^{-5}$), bringing evidence for a bias toward the default option. This constant was significantly reduced when including $DV_{CAT}$ in the regression model ($\beta = 0.31 \pm 0.16$, $t(23) = 1.94$, $p = 0.06$), with the effect of $DV_{CAT}$ itself being significant ($\beta = 0.021 \pm 0.006$, $t(23) = 3.53$, $p = 2.10^{-3}$), which established a direct link between prior preference and default bias. We also introduced past choices (coded 1 vs. −1 when default option was chosen vs. unchosen) in the regression model but they yielded no significant effect on choice rate. Consistently, the constant estimate was not different when restricting the logistic regression to the first choice in a block ($\beta_{first} = 0.58 \pm 0.49$, $\beta_{all} = 0.68 \pm 0.13$, difference: $t(23) = 0.97$ $p = 0.44$), confirming that the default bias was not resulting from the history of past choices. To illustrate this result (*Figure 2A*), we plotted the choice rate, P(def), as a function of the decision value, $DV_{IT}=V_{IT}(def)$-$V_{IT}(alt)$. This plot shows that even when the two options have the same value ($DV_{IT} = 0$ on the x-axis), the choice rate is not at chance level (50% on the y-axis), which would denote indifference, but shifted toward the default option (by $15.7 \pm 1.7$% on average). Thus, these results provide behavioral evidence for a 'choice bias' occurring on top of the decision value ($DV_{IT}$), i.e. above and beyond what could be predicted by the difference in likeability rating.

To account for choice response time (RT), we fitted a general linear model (GLM) including the main effects and the interaction of two factors: the unsigned decision value ($|DV_{IT}|$) and the choice type (default vs. alternative). As typically reported, we found a significant effect of unsigned decision value ($t(23) = -6.8$, $p = 6.10^{-6}$), indicating that choices were longer when option values were closer. We also found a significant effect of choice type ($t(23) = -5.47$, $p = 1.10^{-5}$), indicating that subjects were faster to pick the default option than the alternative. There was no significant interaction between the two factors ($t(23) = 0.59$, $p = 0.56$). Thus the 'RT bias' corresponds to the difference between intercepts for a null decision value (*Figure 2B*). This RT bias means that subjects were significantly faster when choosing the default (by $357 \pm 50$ ms on average), irrespective of the decision value.

To assess whether the choice and RT biases could arise from the same underlying computation, we tested their correlation across blocks (*Figure 2C*). This is possible in our design because each block corresponds to a confrontation between two given categories, some being very close and others far apart in terms of mean likeability (i.e., they vary in terms of $DV_{CAT}$). We fitted a regression model to each block in order to extract choice and RT biases for each pair of categories. Correlation across blocks was estimated at the subject level and then tested against the null hypothesis at the group level. We found a significant correlation between the two biases ($r = 0.24 \pm 0.06$, $t(23) = 3.78$, $p = 1.10^{-3}$), suggesting a common underlying mechanism, which we further characterized using computational modeling.

## Computational modeling

To account for both choice and RT distributions, we employed an analytical approximation to the Drift Diffusion Model (DDM). The DDM assumes that choices result from a sequential sampling process, through which a decision variable accumulates evidence until it reaches a boundary (*Ratcliff, 1978*; *Ratcliff and McKoon, 2008*). DDMs were originally developed to explain perceptual decisions but they have already been successfully applied to economic (value-based) decisions (*Gold and Shadlen, 2007*; *Basten et al., 2010*; *Krajbich et al., 2010*; *2012*). In our DDM, the boundaries corresponded to the default and alternative choices, and the mean of the drift rate was the signed decision value, $DV_{IT}$ (inset in *Figure 2D*). A priori, the choice and RT biases could arise from a change in the drift rate or from a shift in the starting point, S. The latter possibility is more consistent with the negative correlation that was observed between choice bias and RT (*Figure 2D*) and tested at the group level ($r = -0.68 \pm 0.08$, $t(23) = -8.06$, $p = 4.10^{-8}$). Indeed, in the DDM framework, a bias in the starting point has less impact on choices when the decision process lasts longer (*Brunton et al., 2013*).

To formally disentangle between these possibilities, we compared a DDM where the starting point is fixed at zero and the drift rate equal to $DV_{IT}$ (null model) to six alternative DDMs where either the starting point or the drift rate is allowed to change across subjects, and for some of them across blocks. The first three models ('start family') test the hypothesis of a shift in the starting point. The shift was captured with a single free parameter in model 1 ('1 free S'), with one free parameter per block in model 2 ('12 free S'), or as a free parameter scaled by $DV_{CAT}$ in model 3 ('S=a*$DV_{CAT}$').

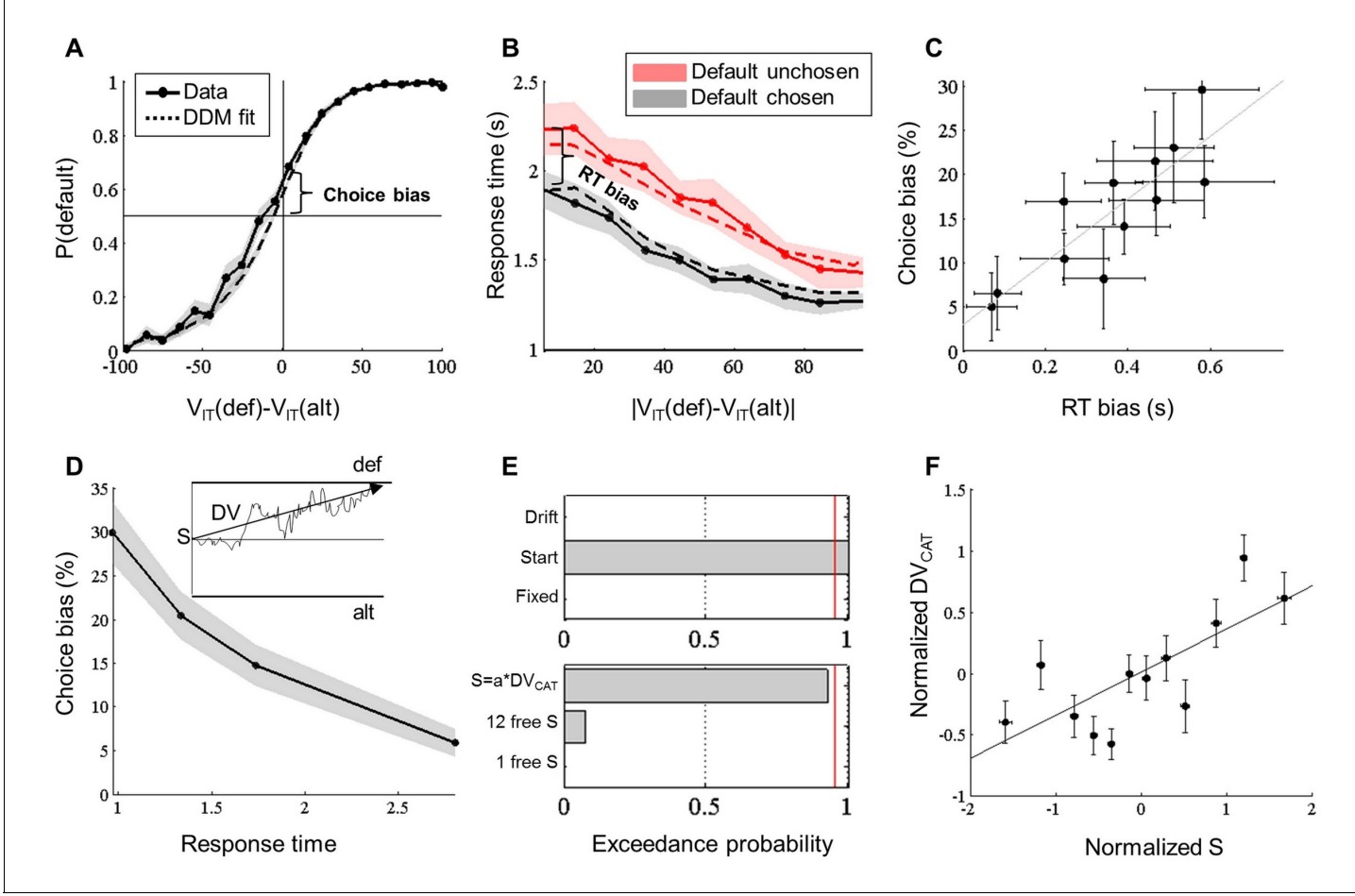

**Figure 2.** Behavioral results (MRI experiment). (A) Probability of choosing the default option, P(def), plotted as a function of decision value, $DV_{IT}$, divided into 20 bins. Values correspond to likeability ratings given by the subject prior to scanning. Both probabilities observed in choice data (solid line) and simulated from the fitted Drift Diffusion Model (dashed line) are shown. Choice bias was defined as the difference between the observed probability for a null decision value and the expected equiprobability (50%). (B) Choice response time (RT) plotted as a function of the absolute decision value, $|DV_{IT}|$, divided into 10 bins, separately for trials in which the default option was chosen (black) and unchosen (red). Both RT observed in behavioral data (solid line) and simulated from the fitted Drift Diffusion Model (dashed line) are shown. RT bias was defined as the difference between the intercepts observed for the two types of choice. (C) Correlation of choice and RT biases across blocks. (D) Choice bias plotted as a function of response time, divided into four bins. Inset illustrates the Drift Diffusion model (adapted from (**Voss et al., 2013**), with S the starting point, DV the mean drift rate and def / alt the thresholds for choosing default / alternative options. Choice bias was larger for shorter RT, suggesting that it could arise from a bias in the starting point. (E) Family model comparison between different theoretical accounts of choice and RT biases. Top: the null model ('Fixed') is compared to models in which either the starting point ('Start') or the drift rate ('Drift') is allowed to favor the default option. Bottom: the model with a single free starting point ('1 free S') is compared to models in which the starting point is varied across blocks, either in proportion to the value difference between categories '$S=a*DV_{CAT}$' or as a set of 12 independent parameters ('12 free S'). Red line corresponds to 95% exceedance probability. (F) Correlation across blocks between $DV_{CAT}$ and starting point S (from fitting the '12-free-S' model). This suggests that the starting point is adjusted in each block to the average value difference between the two confronted categories. Shaded areas and error bars represent ± inter-subject SEM.

Thus, the starting point was respectively considered constant across blocks (but possibly different from zero), freely adjusted to each block, or proportional to the prior preference. The last three models (drift family) test the hypothesis of a change in the drift rate, which in any case was proportional to $DV_{IT}$. The change was captured with a single additional parameter to $DV_{IT}$ in model 4, with one additional parameter per block in model 5, and with an additional term scaled by $DV_{CAT}$ in model 6.

We first conducted a family model comparison to examine the possibilities that the choice and RT biases were due to a shift in the starting point (models 1–3) or a change in the drift rate (models

4–6), relative to the null model (*Figure 2E*, top). The most plausible mechanism was the shift in the starting point (start family: exceedance probability, xp = 0.997). Then, we compared the three models within this family (*Figure 2E*, bottom) and found evidence in favor of model 3 (xp = 0.920), suggesting that the starting point varied across blocks proportionally to prior preferences. We verified this conclusion by testing the correlation across blocks between the posterior means of the 12-free-S model and the prior preference $DV_{CAT}$ (*Figure 2F*). The correlation was significant at the group level (r = 0.35 ± 0.07, t(23) = 5.12, p = $3.10^{-5}$), strengthening the idea that prior preference was imposing a shift in the starting point that resulted in both choice and RT biases. Thus, the correlation observed between choice and RT biases was driven by variations in $DV_{CAT}$ across blocks, the two biases trending to zero when $DV_{CAT}$ was close to null.

## Eye-tracking

The fact that decision bias was best explained by shifting the starting point made less likely an interpretation in terms of attentional dynamics. This is because in previous studies, gaze fixation pattern was found to affect the drift rate and not the starting point (*Krajbich et al., 2010*). We nevertheless investigated the possibility that the effect of prior preferences on choice and RT biases could be mediated by the pattern of gaze fixation. This possibility would imply that subjects pay more attention to the default option than to the alternative, which we examined using eye-tracking measurements.

Another group of participants (n = 23) performed the same series of rating and choice tasks, while their gaze position on the screen was recorded using an eye-tracking device. All the behavioral results described in the previous section were replicated (*Figure 3A and B*), with a significant bias in both choice (15.5 ± 1.7%, t(22) = 5.12, p = $4.10^{-4}$) and RT (341 ± 42 ms, t(22) = −6.69, p = $1.10^{-6}$), and a significant correlation between the two (r = 0.22 ± 0.06, t(22) = 3.87, p = $8.10^{-4}$).

We also replicated a number of results predicted by the attentional Drift Diffusion Model (aDDM), in which a parameter θ down-weights the value of the unattended item in the decision value, hence in the drift rate (*Krajbich et al., 2010*; *Krajbich and Rangel, 2011*; *Lim et al., 2011*; *Krajbich et al., 2012*). As predicted by the aDDM, we notably observed that the choice probability was higher for the item fixated last (t(22) = −11.68, p = $7.10^{-11}$), and for the most fixated item during the decision process (t(22) = −4.71, p = $1.10^{-4}$), irrespective of decision value. These results confirm that fixation pattern had the expected effects on choice. However, none of these effects could account for the bias toward the default option that was observed in our task.

To test the link between prior preference and gaze fixation pattern, we compared the duration of fixation for the default and alternative options, separately for trials in which the default was chosen and unchosen. We found that the default option was fixated longer when it was chosen (difference: 81 ± 11 ms, t(22) = 7.52, p = $2.10^{-7}$). Conversely, it was the alternative option that was fixated longer when the default option was not chosen (difference: 41 ± 17 ms, t(22) = 2.41, p = $2.10^{-2}$). Consistently, the ANOVA conducted on fixation time revealed a significant main effect of the 'chosen vs. unchosen' factor (F(1,88) = 4.03, p = 0.048), but no main effect of the 'default vs. alternative' factor (F(1,88) = 0.43, p = 0.41). The interaction was significant (F(1,88) = 17.45, p = $1.10^{-4}$), reflecting the fact that the default option was more frequently chosen, and with larger decision values. Thus, fixation duration was indeed predictive of choice, but was not influenced by prior preference. To control for the dynamics of the decision process, we computed the proportion of fixations for each option at each time point. The time courses locked to stimulus onset revealed a clear preference for looking at the left option during the first 250 ms, and then at the right option during the next 250 ms, but no significant preference for the default option at the beginning of the trial. The time courses locked on the response confirmed the fixation bias toward the chosen option, with a similar pattern whether default or alternative option was chosen (*Figure 3C*). This result was further confirmed by a model comparison showing that fixation duration for each option was better explained (xp = 0.999) by a GLM including the unsigned decision value and the choice (chosen vs. unchosen option) than by GLMs including an additional regressor that indicated the prior preference (default vs. alternative option).

Finally, we compared four variants of the DDM to contrast how fixation pattern and prior preference influence the decision process. The first was the null model, with a starting point S fixed at zero and a weighting factor θ fixed at one. The second was the sDDM selected as the best model in the first experiment, with S proportional to $DV_{CAT}$ and θ still fixed at one. The third was the standard

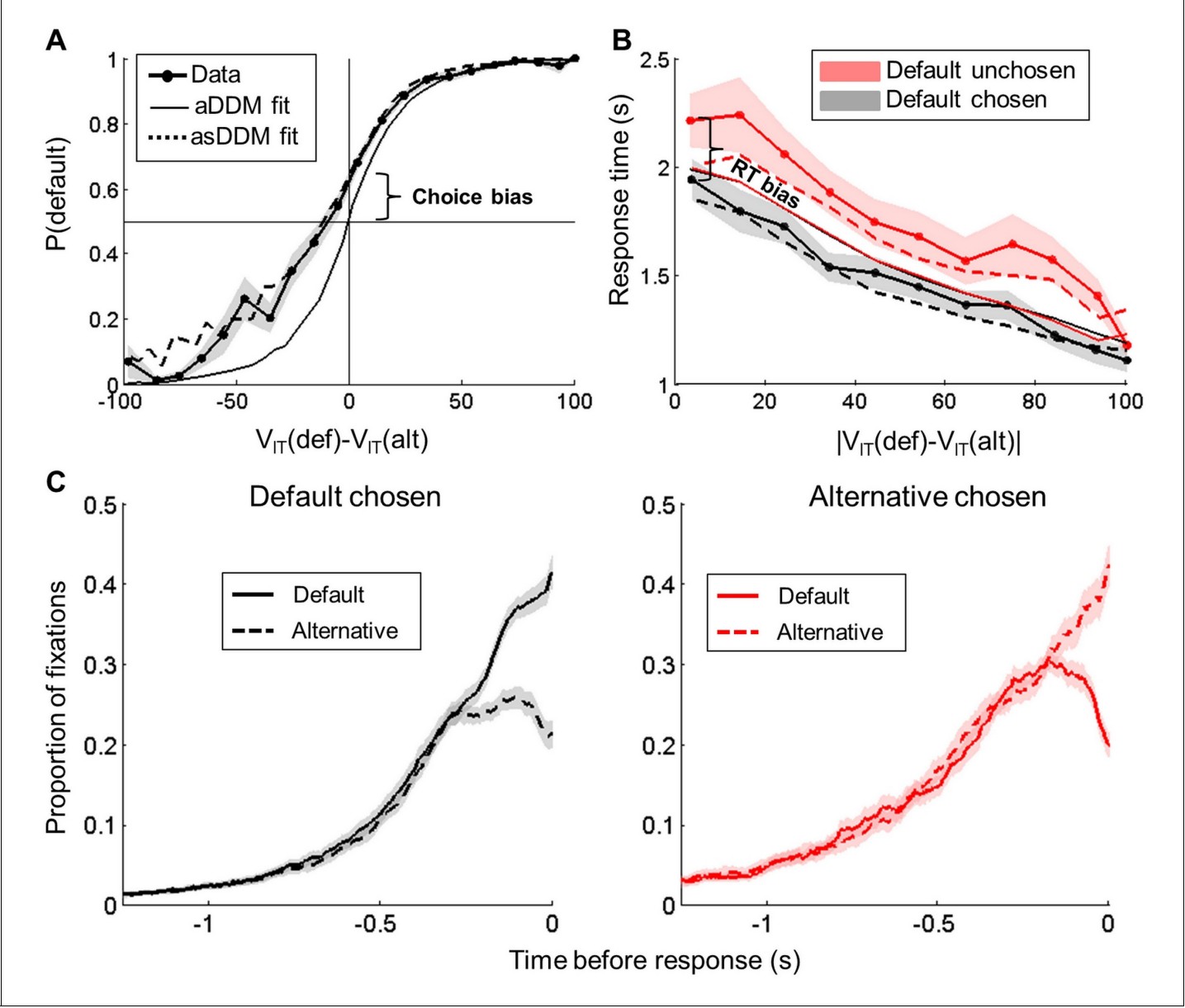

**Figure 3.** Behavioral results (eye-tracking experiment). (**A**) Probability of choosing the default option plotted as a function of decision value DV$_{IT}$. The three curves correspond to probabilities actually observed in choice data (lines with circles) and simulated from either the fitted attentional Drift Diffusion Model (aDDM, solid lines) or the same model fitted with a starting point proportional to prior preference DV$_{CAT}$ (asDDM, dashed line). (**B**) Choice response time (RT) plotted as a function of the absolute decision value |DV$_{IT}$|, separately for trials in which the default option was chosen (left) and unchosen (right). The different curves correspond to RT observed in behavioral data (lines with circles) and simulated from either the fitted aDDM (solid line with circles) or asDDM (dashed line). Note that the aDDM alone cannot reproduce choice and RT biases. (**C**) Proportion of fixations (number of trials over all trials) to the default and alternative options at each time point when default is chosen (left) or unchosen (right). Curves are time-locked to choice (button press). They do not add up to one because at a given time point in a given trial, subjects may fixate none of the two options. Shaded areas are ± inter-subject SEM.

aDDM, with S fixed at zero and a freely fitted θ. The fourth was termed asDDM and included both S proportional to DV$_{CAT}$ and fitted θ. The most plausible model was the asDDM (xp = 0.95), with the weight on DV$_{CAT}$ significantly above zero (0.02 ± 0.003, t(22) = 7.85, p = 1.10$^{-6}$), and a θ significantly below one (θ = 0.94 ± 0.03, t(22) = 2.14 p = 0.04). The fits of choice and RT are illustrated for the aDDM and asDDM (*Figure 3A and B*). Although using the fixation pattern (with θ) improved the fit,

only the prior preference (with S) could explain the decision bias toward the default option. We reached similar conclusions when the advantage for the attended option was additively included in the drift rate, on top of decision value (as in *Cavanagh et al., 2014*). In fact, gaze fixation pattern failed to produce the default bias simply because the default option was no more looked at than the alternative option.

## fMRI

Our behavioral results establish that prior preferences exert a bias on choices, which in the DDM framework was best explained by a proportional shift in the starting point. We analyzed fMRI data first to examine whether the bias toward the default option could be observed in baseline neural activity, second to assess whether prior preference could frame the comparison between option values that might be implemented in the evoked neural response.

### Baseline activity

To examine whether the prior preference ($DV_{CAT}$) was encoded in baseline activity, we fitted a GLM (GLM0, see Materials and methods) convolved with a finite impulse response (FIR) function to fMRI data. GLM0 contained an indicator delta function for option display that was parametrically modulated by the option values, $V_{IT}(def)$ and $V_{IT}(alt)$. In the following we analyze regression coefficient estimates obtained for the indicator function on volumes acquired before and after option display. The contrast performed at the individual level weighted block-specific indicator functions by z-scored $DV_{CAT}$. Group-level statistical test ($p < 0.005$, uncorrected) performed on this contrast for the volume acquired two seconds before option display revealed activity scaling with $DV_{CAT}$ in the vmPFC, ventral striatum and left hippocampus (*Figure 4A*, left panel), which are regions classically identified as parts of the brain valuation system (e.g., *Lebreton et al., 2009*). The ventral striatum and left hippocampus were the only regions that survived cluster-level family-wise error (FWE) correction at the whole-brain level. The vmPFC cluster only survived small-volume correction ($p = 6.10^{-3}$) within an ROI based on independent criterion – a sphere centered on the peak of the cluster that positively reflected value in a previous meta-analysis (*Bartra et al., 2013*). In order to illustrate the time course of this effect in the vmPFC, we simply averaged BOLD activity levels (coefficient estimates for indicator functions) in high and low $DV_{CAT}$ blocks separated with a median split (*Figure 4A*, right panel). The difference between high and low $DV_{CAT}$ appeared to be maintained during the decision process and to progressively vanish at the end of the trial.

### Evoked response

Our key hypothesis was that the decision value signal is framed by the prior preference, as a comparison between default and alternative options. This opposition of default vs. alternative options partially overlaps with that of chosen vs. unchosen options (here, in 77.8 ± 1.0% of the choices), the latter contrast being classically used to localize value comparison signals. We started the analysis by replicating this classical approach with a standard GLM, before dissociating the two possible frames with a more exhaustive GLM.

The first GLM only contained $V_{IT}(ch)$ and $V_{IT}(unch)$ as parametric modulators of a categorical regressor (delta function) indicating option display (GLM1, *Figure 4—figure supplement 1A*, left), all convolved with a canonical hemodynamic response function (HRF). As expected, the classical contrast $V_{IT}(ch)-V_{IT}(unch)$ revealed significant correlation in brain valuation regions such as the vmPFC, ventral striatum and posterior cingulate cortex. Yet, this pattern of activation was not very specific of the brain valuation system, as it also included other brain regions such as the intra-parietal lobules. The opposite contrast, $V_{IT}(unch)-V_{IT}(ch)$, yielded significant correlation in the dorsal anterior cingulate cortex (dACC) and bilateral anterior insula, which are classically associated with choice difficulty or with the value of foregone options (*Kolling et al., 2012*; *Shenhav et al., 2014*), as well as in the inferior frontal gyrus and middle occipital gyrus. The equivalent GLM containing $V_{IT}(def)$ and $V_{IT}(alt)$ as parametric modulators yielded similar results (GLM1', *Figure 4—figure supplement 1A*), which is expected due to the shared variance between the chosen and default option values.

In order to disambiguate between representations of pre-choice values, $V_{IT}(def)$ and $V_{IT}(alt)$, and post-choice values, $V_{IT}(ch)$ and $V_{IT}(unch)$, we built a second GLM (GLM2, *Figure 4—figure supplement 1B*, left) that included two value regressors for each choice type (default chosen versus

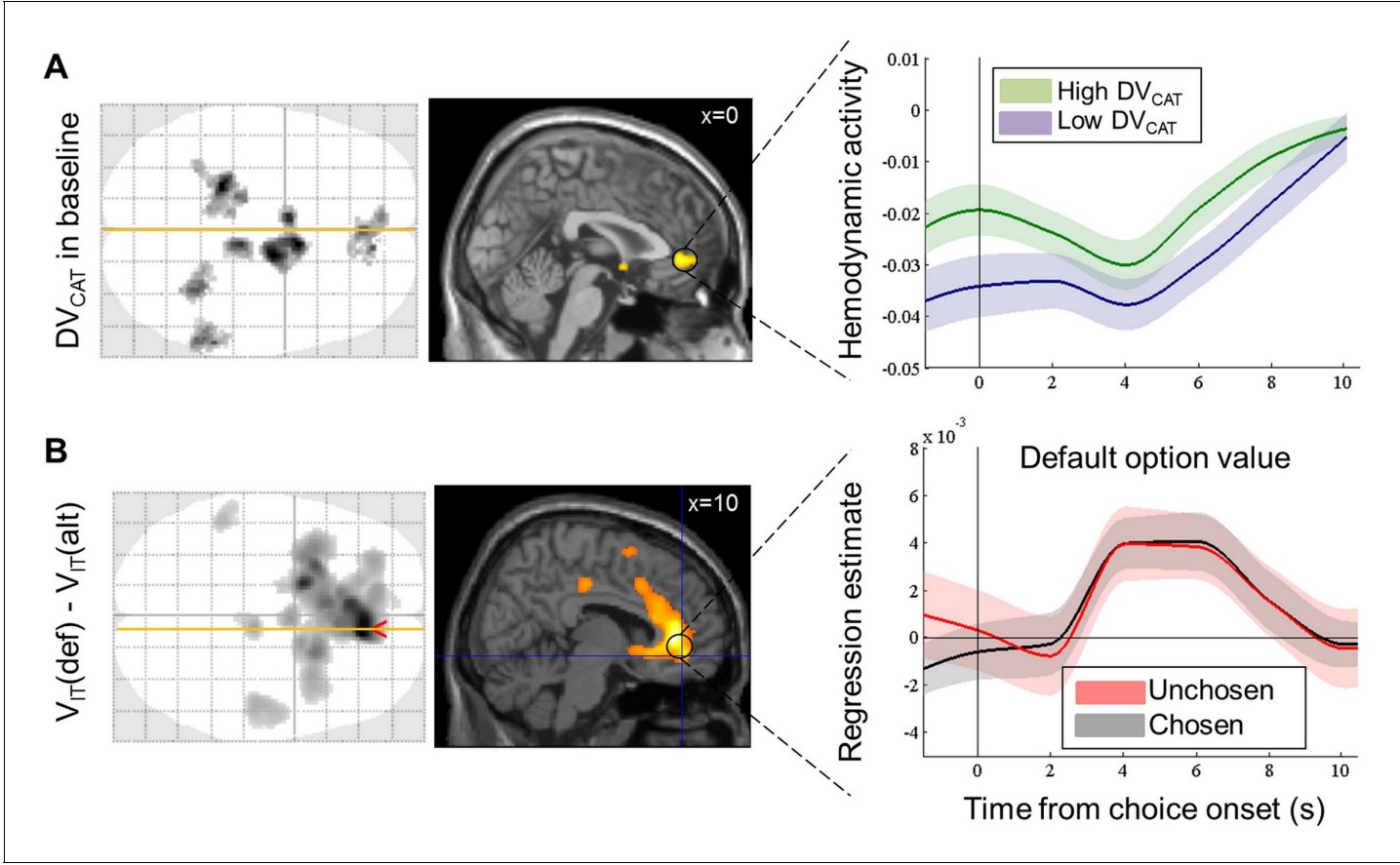

**Figure 4.** Neural correlates of the bias and framing effects of prior preference. (**A**) Bias in value coding and vmPFC baseline. Left: axial glass brain and sagittal slice of statistical maps relating to the prior preference (DV_CAT), one volume (2 s) before the display of choice options. Areas shown in black (on glass brain) and yellow (on sagittal slice) showed significant positive group-level random effect (one-sample t-test, p<0.005 uncorrected for display purposes, minimum extent: 100 voxels). Right: time course of peri-stimulus fMRI activity in the vmPFC region, shown separately for high (green) and low (purple) DV_CAT. Black vertical line (time 0) indicates the onset of choice options. (**B**) Frame of value coding in vmPFC response. Left: axial glass brain and sagittal slice of statistical maps relating the canonical hemodynamic response to the decision value (DV_IT), at the onset of choice option (same threshold as in **A**: one-sample t-test, p<0.005 uncorrected for display purposes, minimum extent: 100 voxels). Right: time courses of the regression estimate (beta) obtained in the vmPFC region for the default option value when it was chosen (black) or unchosen (red). Shaded areas are ± inter-subject SEM.

The following figure supplement is available for figure 4:

**Figure supplement 1.** Dissociation of neural value representations expressed in the pre-choice versus post-choice frame.

unchosen). These value regressors were parametric modulators of the categorical regressor indicating option display. The GLM also contained a regressor modeling the choice type itself, to dissociate value coding from option selection, and a boxcar function parametrically modulated by DV_CAT, to account for tonic effects of prior preference. This GLM allows computing both the V_IT(ch)-V_IT(unch) and the V_IT(def)-V_IT(alt) contrasts on the canonical evoked response.

Critically, we found significant activation (surviving cluster-level, whole-brain FWE correction) in the vmPFC and ventral striatum with the V_IT(def)-V_IT(alt) contrast (*Figure 4B*, left) but not with the V_IT(ch)-V_IT(unch) contrast (*Figure 4—figure supplement 1B*, right). The neural response implementing the V_IT(def)-V_IT(alt) comparison was specific to the brain valuation system, since no other regions than vmPFC and ventral striatum passed the corrected threshold (Table S1 - *Supplementary file 1*). In particular, activity in the parietal or temporo-parietal cortex followed the V_IT(ch)-V_IT(unch) contrast as in the classical GLM. The opposite contrast, V_IT(unch)-V_IT(ch), again activated the dorsal anterior cingulate, anterior insula and middle occipital gyrus (*Figure 4—figure supplement 1B*, right). The

latter activation might relate to the fact that visual inspection of choice options was longer when the choice was more difficult. No brain region was significantly associated with the $V_{IT}(alt)-V_{IT}(def)$ contrast.

As we realized that the evoked response might have contaminated baseline vmPFC activity at the next trial, we estimated another version of GLM2, with the four parametric modulators - $V_{IT}(def\_chosen)$, $V_{IT}(alt\_unchosen)$, $V_{IT}(def\_unchosen)$, $V_{IT}(alt\_chosen)$ – replaced by the same values but for the options presented in the previous trial (the first trial of each block was discarded). None of these parametric modulators had a significant effect at the time points preceding option display (t = −2 and t = 0 s from stimulus onset). Therefore, the values of the options presented in the previous trial did not affect baseline vmPFC activity, beyond the variance that they shared with $DV_{CAT}$.

Thus, the neural decision value encoded in the vmPFC seemed to be expressed in the pre-choice frame (default minus alternative), rather than in the post-choice frame (chosen minus unchosen). However, inspecting the regression coefficient estimates obtained for $V_{IT}(def)$ and $V_{IT}(alt)$ separately suggested that the contrast was driven by $V_{IT}(def)$, as no significant effect was observed with $V_{IT}(alt)$ alone. To verify that $V_{IT}(def)$ was similarly encoded in the vmPFC irrespective of the eventual choice, we fitted a FIR version of GLM2, and extracted regression estimates from the same independent vmPFC ROI as used previously (*Bartra et al., 2013*). We found that regression estimates for $V_{IT}(def)$ were significant for both choice types (default chosen: $\beta = 6.10^{-3} \pm 2.10^{-3}$, t = 3.23, p = $3.10^{-3}$; default unchosen: $\beta = 6.10^{-3} \pm 3.10^{-3}$, t = 2.67, p = 0.01), four seconds after option display. For illustration purposes, we have plotted the time course of regression estimates extracted from the vmPFC cluster associated with $V_{IT}(def)-V_{IT}(alt)$ in our main analysis (*Figure 4B*, left).

To further challenge our conclusion regarding the encoding of decision value in the vmPFC, we compared variants of GLM2 that included different parametric regressors locked to option display. The two-by-two model space tested the possibilities of (1) pre-choice (default minus alternative) versus post-choice (chosen minus unchosen) framing for value coding and (2) best option value ($V_{IT}(def)$ or $V_{IT}(ch)$) versus differential value coding ($DV_{IT}$). Bayesian model selection indicated that the pre-choice family was more plausible (xp = 0.92), and that within this family, $V_{IT}(def)$ alone better accounted for vmPFC activity than $DV_{IT}$ (xp = 0.94). By comparison, the post-choice family best accounted for dACC activity (xp = 0.99), with a clear advantage for the decision value $DV_{IT}$ over $V_{IT}(ch)$ alone (xp = 0.9996).

At first, we were surprised that the vmPFC (or any other brain region) seemed to hold no representation of $V_{IT}(alt)$ despite the fact that, behaviorally, $V_{IT}(alt)$ impacted choices to a similar extent as $V_{IT}(def)$. Then we reasoned that the vmPFC might have encoded both option values on top of the decision value, following on the idea that such valuation processes are automatically triggered when stimuli are presented (*Lebreton et al., 2009*; *Levy et al., 2011*). This would imply that vmPFC activity should correlate not only with the difference between option values but also with their sum. The net result would be a correlation with $V_{IT}(def)$, since $V_{IT}(alt)$ would be subtracted out, which was observed in our analysis. In order to test this hypothesis, we fitted a last variant of GLM2 that included both the difference and sum of option values in the pre-choice frame as parametric modulators of option display. The common variance, linked to the presence of $V_{IT}(def)$ in both the sum and difference, was removed such that each regressor had unique variance. We examined the regression estimates extracted from the independent vmPFC ROI and found significant effects for both the difference ($\beta_{def-alt} = 0.10 \pm 0.04$, t(23) = 2.74, p = 0.01) and the sum ($\beta_{def+alt} = 0.14 \pm 0.06$, t(23) = 2.47, p = 0.02). In addition, model comparison showed that the GLM with orthogonalized sum and difference was a much better explanation of the vmPFC response (xp = 0.993) than the GLM containing only $V_{IT}(def)$. This result suggests that the vmPFC evoked response signals the two option values (sum) on top of the decision value (difference). In macroscopic measures of brain activity such as fMRI, the positive and negative correlation with $V_{IT}(alt)$ appear to cancel each other, but these representations might be dissociable using techniques with better spatial resolution that can access microscopic scales.

## Discussion

In this study, we examined how prior preference shapes the neural representation of decision value. We observed two major phenomena in vmPFC activity: (1) baseline activity was shifted in proportion to the strength of prior preference, as was the starting point in a drift diffusion model accounting for

the decision bias in favor of the default option, (2) evoked activity signaled the value of the option belonging to the preferred category, suggesting that the choice was framed as a comparison between default and alternative.

Although they were not instructed to do so, subjects likely formed prior preferences at the beginning of blocks, when the two categories confronted in the upcoming series of choices were announced on the screen. Preference between the two categories was inferred from likeability ratings averaged across items belonging to each category. In a vast majority of cases, this notion of preference matched the preference directly expressed by the subjects in binary choices between categories made during post-scan debriefing. Moreover, the difference between mean likeability ratings ($DV_{CAT}$) was proportional to the confidence expressed in these choices between categories, in keeping with the notion that choice and confidence proceed from the same decision value (*De Martino et al., 2013*). These debriefing observations validate our notion of prior preference, which then served to designate the default option in the pair of items that was presented for choice.

Indeed items from the preferred category could be qualified as default options, because they were chosen faster and more frequently than their alternatives. These choices and RT biases are minimal requirements for a default option, i.e. an option that should be chosen in the absence of further information processing. Such criteria have been used in other paradigms where the goal was to maximize an objective reward, with for instance the default being defined as the pre-selected option in a perceptual decision task (*Fleming et al., 2010*), as the current patch in a foraging task (*Hayden et al., 2011*; *Kolling et al., 2012*; *Kvitsiani et al., 2013*), or as the long-term best option in a probabilistic instrumental learning task (*Boorman et al., 2013*). These studies reported that when the two option values were similar, subjects (both humans and monkeys) favored the default option. This phenomenon has been coined 'default bias', or 'status quo bias' in cases where the default option was defined as the pre-selected choice. Here, the same phenomenon was observed in the case of subjective preference. Importantly, the default bias was estimated once option values were matched, therefore it goes beyond what could be predicted from the difference in likeability ratings between categories. This bias could lead to preference reversals, meaning that subjects would pick the default option in spite of the alternative option having received a higher rating. Thus, the default bias denotes suboptimal decision-making, which could be compensated by the fact that following a default policy is on average less costly in terms of time or cognitive resource, than a systematic unbiased comparison of option values. This phenomenon is therefore much different from the optimal use of prior information that has been observed in a variety of perceptual decision-making paradigms, subjects being biased only when tricked with invalid cues (*Link and Heath, 1975*; *Bogacz et al., 2006*; *Scheibe et al., 2010*; *Mulder et al., 2012*).

Within the drift diffusion framework, the default bias observed in choice and RT was best accounted for by a shift in the starting point. This is consistent with perceptual decision-making studies showing that prior information on probability or payoff is also incorporated in the starting point (*Scheibe et al., 2010*; *Summerfield and Koechlin, 2010*; *Mulder et al., 2012*). However, this is not compatible with the idea that the effect of prior preference on choice and RT biases is mediated by the pattern of gaze fixations. This idea implies that subjects pay more attention to the default option, which through the attentional DDM mechanism would favor the default choice, because the attended option has more weight than the alternative in the drift rate. In fact, subjects looked equally often at the default and alternative options in the eye-tracking experiment. Our results nonetheless confirmed that the pattern of gaze fixation does inform the prediction of choices, in a manner that is nicely captured by the attentional DDM. Thus, although the attentional DDM is perfectly compatible with our data, it could not by itself explain the default bias. The best account of choice and RT was in fact obtained with a model that cumulated the down-weighting of unattended options in the drift rate, as suggested by the attentional DDM, and the shift in the starting point that explains the default bias.

In our best model, the shift in starting point was proportional to the prior preference ($DV_{CAT}$). A striking parallel was found at the neural level, with the prior preference being reflected in the baseline activity of valuation regions including the vmPFC, ventral striatum and posterior cingulate cortex. This is in line with a previous study showing that baseline vmPFC activity is sensitive to contextual factors, both in humans and monkeys, and provide a bias in subsequent valuation processes (*Abitbol et al., 2015*). Other contextual manipulations were found to modulate vmPFC activity and subsequent valuation, for instance mood induction has been shown to affect reward-related

vmPFC activity (*Young and Nusslock, 2016*). In contrast, cueing manipulation that affected perceptual decisions through a shift in starting point had no influence on vmPFC activity (*Scheibe et al., 2010*; *Summerfield and Koechlin, 2010*; *Mulder et al., 2012*). This dissociation suggests that the recruitment of vmPFC was not related to the general process of changing the accumulation starting point, but to biasing value-based decisions (as opposed to perceptual decisions). In fact the shift in baseline vmPFC activity was maintained throughout the decision process, and was hence added to the evoked activity, which followed a canonical hemodynamic response. As both baseline and evoked activity scaled with the value of the default, respectively at the category and item levels, they together contributed to favoring the default option over the alternative. Thus, the mechanics is analog to the DDM process, but the dynamics is somewhat different. In fact, the neural dynamics is not compatible with the vmPFC implementing the DDM, since we observed no ramping signal corresponding to an accumulation-to-bound process; neither is it compatible with the vmPFC output being sent to a distant accumulator, since the shift in starting point should not be integrated over time. Therefore, we do not suggest that the DDM used to capture behavioral patterns is literally implemented as such in the brain, just that the general logics and some key features appeared to match vmPFC activity during choices. We also note that other types of modeling would have been possible to capture behavioral effects, notably a Bayesian account where prior preference would affect the mean and perhaps the variance of a prior distribution on decision value.

The analysis of the evoked response showed that the vmPFC and ventral striatum encode the decision value in a frame that opposes the default to the alternative option. This pre-choice framing supports the idea of an anatomical separation between the valuation and selection processes, with the vmPFC being implicated in the former but not the latter. It could be a very general frame for value coding in the vmPFC, because most studies found a correlation between vmPFC activity and the value of chosen options (e.g., *Hare et al., 2011*; *Boorman et al., 2013*), which are partially confounded with default options as we have shown here. We note that an opposite dissociation was found by *Boorman et al. (2013)*, with the vmPFC encoding option values in post-choice frame, and not pre-choice frame. As decision-making dynamics was not explored in this study, it is unclear whether participants truly implemented a default strategy as defined here, which implies an anticipation of a default response, associated with shortening of response time. Accordingly, the representation of chosen option value was largely delayed in comparison to our study (peaking 10 s after option display), possibly related to the necessity of storing expected values in a learning context.

Another partial confound is with choice easiness or confidence, which was also found to be integrated in vmPFC activity in addition to value (*De Martino et al., 2013*; *Lebreton et al., 2015*). The pre-choice framing could also be reconciled with the theory that the vmPFC encodes the value of the attended option, if we assume that when they have no prior information on the choice, subjects set up a default on the fly, which could be the option they just look at. By contrast, we found a post-choice framing of decision value (unchosen vs. chosen) in the dACC and anterior insula, which could be related either to choice difficulty or to the value of shifting away from the default policy, which might require cognitive control (*Hare et al., 2011*; *Kolling et al., 2012*; *Shenhav et al., 2013*).

A last potential issue is that the correlation with decision value ($DV_{IT}$) was driven by the default option, although the default and alternative options had the same weight on choices, and despite the two options being reflected in other regions such as dACC. Our interpretation is that both option values are encoded in the vmPFC on top of the decision value. As a result, the correlation with the alternative option value would be cancelled out, and the correlation with the default option value would be doubled, as suggested by the following equation:

$$Signal(vmPFC) = [V_{IT}(def) - V_{IT}(alt)] + [V_{IT}(def) + V_{IT}(alt)] = 2 * V_{IT}(def)$$

This interpretation is consistent with both the idea that the vmPFC automatically encodes the value of items that fall under the attentional focus (*Lebreton et al., 2009*; *Levy et al., 2011*) and the idea that the vmPFC computes a decision value whenever a choice process is engaged (*Plassmann et al., 2007*; *Grueschow et al., 2015*). It would also explain why many studies report a correlation with the chosen value alone and not the decision value, as the unchosen value would be cancelled out for the same reasons (*Wunderlich et al., 2010*; *Kolling et al., 2012*; *Hunt et al., 2012*). Model comparison supported this post-hoc interpretation, as including the two option values (sum) on top of the decision value (difference) provided the best account of vmPFC activity during

choice. Other techniques than fMRI, with better spatial resolution, would be needed to investigate whether the different value representations rely on different populations on neurons.

In conclusion, our findings show that when decision-makers have a prior preference, the brain valuation system is configured so as to compare default and alternative options, with prior and novel information being encoded in baseline and evoked activity, respectively. Such framing could have been selected to solve natural decision problems, with the advantage of saving time and/or cognitive resource, and the disadvantage of biasing choice toward the default policy. How the valuation system adapts to artificial economic choices, in which two novel options present themselves simultaneously, still needs to be investigated. One may speculate that the brain would start by defining a default option, and then proceed to the comparison as usual. If this is correct, identifying the trial-wise and/or subject-wise default policy might be essential for understanding how the brain makes value-based decisions. However, we only have a proof of concept here, the generality of the 'default vs. alternative framing' remains to be established. Further research is also required to specify the contribution of the different brain regions that are involved in the valuation and selection processes, notably the dACC. The present results suggest that the vmPFC provides a decision value, which is also represented in the ventral striatum. How such decision value is used by the brain to make a selection remains to be explained.

## Materials and methods

### Participants

The study was approved by the Pitié-Salpétrière Hospital ethics committee. All subjects were recruited via e-mail within an academic database and gave informed consent before participation in the study. They were right-handed, between 20 and 32 years old, with normal vision, no history of neurological or psychiatric disease, and no contra-indication to MRI (pregnancy, claustrophobia, metallic implants). They were not informed during recruitment that they would win food items, music CD and magazines to avoid biasing the sample. In total, 24 subjects (23.8 ± 2.8 years old, 12 females) were included in the fMRI experiment and paid a fixed amount (80€) for their participation. In the eye-tracking experiment, 24 right-handed subjects (24 ± 3.4 years old, 11 females) were recruited following the same procedure with the same inclusion criteria. No statistical method was used to predetermine sample size, but our sample size is similar to those generally employed in the field. One subject was excluded due to a technical issue with the eye-tracking device.

### Tasks

All tasks were programmed on a PC in MATLAB language, using the Psychophysics Toolbox extensions (RRID:SCR_002881, *Brainard, 1997*; *Pelli, 1997*). Subjects performed the rating task outside the scanner and the choice task during fMRI scanning. Prior to each task, they were instructed and trained on short versions (24 trials) to get familiarized with the range of items and the mode of response.

During the rating task, subjects were asked to estimate the likeability of all 432 items that they could potentially obtain at the end of the experiment. These items were blocked by reward domain: food, music and magazines. Unbeknown to subjects, each reward domain was divided into 4 categories of 36 items. The 12 categories were: appetizers, biscuits, candies, chocolate (food domain); news, comics, cultural, generalist (magazine domain); French, jazz, rock, urban (music domain). The order of presentation was randomized within each reward domain, such that the categories were intermingled. The series of trials consisted of displaying pictures of the items one by one on the computer screen. A pseudo-continuous rating scale (101 points) was presented below the picture, with three reference graduations (do not like at all, neutral, like a lot). Subjects could move a cursor along the scale by pressing a key with the right index finger to go left or another key with the right middle finger to go right. The cursor was initially positioned at the middle of the rating scale. The rating was self-paced and subjects had to press a button with the left hand to validate their response and proceed to the next trial. At the beginning of each block, the reward domain was announced on a black screen.

Likeability ratings were used for pairing options in the choice task. For each domain, mean ratings were used to rank categories according to subjective preference. The most preferred categories

(ranked 1 and 2) were opposed to the least preferred ones (ranked 3 and 4), making a total of 4 oppositions (1–3, 1–4, 2–3, 2–4). To generate the series of choices for each opposition, items were sorted in the order of likeability rating. Half the choices varied the difference between ratings while keeping the average constant (item ranked mean+X was paired with item ranked mean-X); the other half varied the average while keeping the difference minimal (item ranked X was paired with item ranked X-1). Thus, the mean value and relative value of choice options were orthogonalized. A total of 36 choices were generated for each inter-categorical opposition, and presented in a randomized order. The 36 choices were divided into 4 blocks of 9 trials, presented in 4 different fMRI sessions. As there were 12 possible oppositions (4 per domain), this makes a total of 432 trials, meaning that each item being presented twice.

At the beginning of each block, the domain was announced on a black screen for 0.5 s, then the two opposed categories were displayed for 2 to 5 s, followed by a 0.5 s fixation cross. Each block was composed of a series of 9 choices. Choice trials started with the display of the two options side by side. The side of a given category as well as the best rated option was counter-balanced across trials. Subjects were asked to indicate their preference by pressing one of two buttons, with their left or right index finger, corresponding to the left and right options. The chosen picture was framed with a white square for 0.5 s, followed by a black screen with fixation cross lasting for 0.5 to 6 s.

Importantly, subjects were not asked to generate a prior preference at the beginning of blocks, when the categories are revealed. They were only told that contextual information would be given before each series of choices, and that it would not require any response from their part. They also knew that at the end of the experiment, one trial per domain would be randomly selected and that they would be given the options chosen in these trials.

Following the scanning session, subjects had to complete a debriefing task in which they were presented the opposed categories two by two. They were asked to first select the category that they preferred and then to rate their confidence in their choice using an analog scale. Finally, they spent an additional 20 min in the lab to eat the food item, listen to the music album and read the magazine that they just won.

## Behavior

All analyses were performed with Matlab Statistical Toolbox (Matlab R20013b, The MathWorks, Inc., USA). Two dependent variables were recorded: choice (which option was selected) and response time (between option onset and button press). The influence of likeability ratings on these variables was assessed with logistic or linear regression models, as explained in the results section. Regression estimates were computed at the individual level and tested for significance at the group level using one-sample two-tailed t-test. Correlations between variables of interest were also computed at the individual level, using Pearson's coefficient, and similarly tested at the group level.

## Eye-tracking

In the eye-tracking experiment, gaze position was recorded with a 60 Hz sampling frequency using The Eye Tribe device (http://theeyetribe.com), during each block of the choice task. There was no constraint on the head, subjects were simply asked to avoid head movement. A screen providing feedback on the eye position was inserted in the trial sequence every time gaze was lost. The number of excluded trials due to loss of gaze position varied between 0 and 6, depending on subjects.

Fixation duration was computed for each trial and option, as the time during which gaze position was inside a square window delineated the corresponding picture on the screen. Last fixation was defined as the picture being looked at when the choice was made. The proportion of fixation was calculated as the number of trials in which gaze position was on the corresponding picture at a given time point. Note that these proportions for the two options do not add up to one because the gaze position can be outside the two windows.

## Modeling

We used the EZ2 analytical approximation for the drift diffusion model (*Wagenmakers et al., 2007*; *Grasman et al., 2009*) to account for the probability of choosing the default option and the response time, on a trial-by-trial basis. As proposed by (*Ratcliff, 1978*; *Wagenmakers et al., 2007*; *Grasman et al., 2009*), we defined the probability of choosing the default option as:

$$P(def) = \frac{\varphi(-A, S-A)}{e^{\frac{2\mu A}{\sigma^2}} - 1}$$

As proposed by EZ2 (*Grasman et al., 2009*), we defined the corresponding RT as:

$$RT(def\ chosen) = Tnd + \frac{(A-S)*(\varphi(S,A) + \varphi(0,A-S) + 2A\varphi(A-S,0))}{-\mu\varphi(A-S,A)\varphi(-A,0)}$$

With $\varphi(x,y) = e^{\frac{-2\mu y}{\sigma^2}} - e^{\frac{-2\mu x}{\sigma^2}}$, $A$ the amplitude between boundaries, $S$ the starting point, $\mu$ the mean of the drift rate, $\sigma$ the standard deviation of the drift rate and $Tnd$ the non-decision time. To compute response time in trials where the alternative option is chosen, we replace $(\mu, S)$ by $(-\mu, A-S)$.

The free parameters $A$, $Tnd$, $\sigma$, $\mu$ and $S$ were estimated with the behavioral data. Different versions of the model were compared to account for the changes in choice and RT patterns that were induced across blocks by the variations in prior preference. In all cases, $A$, $Tnd$ and $\sigma$, were estimated for each individual but constant across blocks. In the null model, $\mu$ was proportional to the decision value (difference in likeability rating between options, $DV_{IT}$, such that $\mu = \alpha DV_{IT}$) and $S$ was set to zero. The model space (see details in the results section) explored the possibilities that $\mu$ and $S$ could differ from their initial setting ($\mu = \alpha DV_{IT} + \beta$ / $S=z$), vary across blocks (12 free $\alpha$ for $\mu$/12 free $z$ for S), or be informed by the prior preference (difference in mean likeability rating between categories, $DV_{CAT}$, such that $\mu = \alpha DV_{IT} + \beta DV_{CAT}$ / $S = \beta DV_{CAT}$). In the attentional versions of the model, $\mu$ was also informed by gaze fixations, as follows:

$$\mu = \frac{(V_{IT}(def) - \theta V_{IT}(alt)) * D_{def} - (V_{IT}(alt) - \theta V_{IT}(def)) * D_{alt}}{D_{def} + D_{alt}}$$

With $D_{def}$ and $D_{alt}$ the total durations of fixation for the default and the alternative options during the considered trial, and $\theta$ the weight discounting the value of the unfixated item relative to the fixated one (*Krajbich et al., 2010*).

All versions of the drift diffusion model were fitted separately for each individual to choices and RTs using Matlab VBA-toolbox (available at http://mbb-team.github.io/VBA-toolbox/), which implements Variational Bayesian analysis under the Laplace approximation (*Daunizeau et al., 2014*). This iterative algorithm provides a free-energy approximation for the model evidence, which represents a natural trade-off between model accuracy (goodness of fit) and complexity (degrees of freedom) (*Friston et al., 2007*; *Penny, 2012*). Additionally the algorithm provides an estimate of the posterior density over the model free parameters, starting with Gaussian priors. Individual log model evidences were then taken to group-level random-effect Bayesian model selection (BMS) procedure (*Penny et al., 2010*). BMS provide an exceedance probability (xp) that measures how likely it is that a given model (or family of models) is more frequently implemented, relative to all the others considered in the model space, in the population from which participants were drawn (*Rigoux et al., 2014*; *Stephan et al., 2009*).

## fMRI

Functional echo-planar images (EPIs) were acquired with a T2*-weighted contrast on a 3 T magnetic resonance scanner (Siemens Trio). Interleaved 2 mm slices separated by a 1.5 mm gap and oriented along a 30° tilted plane were acquired to cover the whole brain with a repetition time of 2.01 s. The first five scans were discarded to allow for equilibration effects. All analyses were performed using statistical parametric mapping (SPM8, RRID:SCR_007037) environment (Wellcome Trust Center for NeuroImaging, London, UK). Structural T1-weighted images were coregistered with the mean EPI, segmented, and normalized to the standard Montreal Neurological Institute (MNI) T1 template. Normalized T1-images were averaged across subjects to localize group-level functional activations by superimposition. During preprocessing, EPIs were spatially realigned, normalized (using the same transformation as for structural images), and smoothed with an 8 mm full-width at half-maximum Gaussian kernel.

We used four general linear models (GLMs) to explain pre-processed time-series at the individual level.

The first model (GLM0) was built for whole-brain search of voxels encoding prior preference in baseline activity. It was composed of a finite impulse response function (FIR) that included seven time points per trial, from one TR ($-2.01$ s) before to five TR (10.05 s) after choice onset. The different blocks were modeled in separate regressors, each being parametrically modulated by the two option values (default and alternative). For each time point, we computed a contrast that weighted all trials of a given block by the corresponding prior preference ($DV_{CAT}$). Four subjects were excluded from this analysis because they presented at least one block without a sufficient variance to estimate the parametric regression coefficients.

The second model (GLM1) included a stick function capturing option display (only one event per trial), parametrically modulated by the two option values (chosen and unchosen). The three regressors were convolved with a canonical hemodynamic response function.

The third model (GLM2) included two categorical regressors: a boxcar function over blocks and the same stick function as in GLM1. The boxcar function was parametrically modulated by $DV_{CAT}$, to account for tonic effects of prior preference. The stick function was parametrically modulated by five variables: chosen option (default or alternative), $V_{IT}(def)$ when default chosen, $V_{IT}(alt)$ when default chosen, $V_{IT}(def)$ when default unchosen and $V_{IT}(alt)$ when default unchosen. This allowed computing orthogonal contrasts for the decision value in the pre-choice (default vs. alternative) and post-choice (chosen vs. unchosen) frames. The regressors were convolved with a canonical HRF to localize brain regions where the evoked response reflected the decision value. In a subsequent analysis the same regressors were convolved with the same FIR as used for GLM0, in order to examine the dynamics of value coding in regions of interest (ROI).

The fourth model (GLM3) was equivalent to GLM1 except that the stick function modeling option display was modulated by the sum and difference of option values, in the pre-choice frame (default vs. alternative). Common variance between the two parametric regressors was removed such that they could explain a unique variance in the BOLD signal.

Motion artifacts were corrected in all GLMs by adding subject-specific realignment parameters as covariates of no interest. Regression coefficients were estimated at the individual level and then taken to group-level random-effect analysis using one-sample two-tailed t-test. In ROI analyses they were extracted from spheres of 6 mm radius positioned on coordinates defined independently from the present dataset: for the vmPFC we took the peak coordinates [$-2$ 40 $-8$] from a meta-analysis of value representation (*Bartra et al., 2013*), and for the dACC we took the peak coordinate [$-6$ 24 34] of a negative correlation with chosen option value (*Boorman et al., 2013*).

Four variants of GLM2 were also compared to better characterize value coding in the two ROI. The four regressors modeling option values were replaced by a single regressor: (1) default option value, (2) pre-choice decision value (default minus alternative), (3) chosen option value, (4) post-choice decision value (chosen minus unchosen). This was meant to assess whether value representation concerned only one option or the difference between the two, and whether it was expressed in a pre-choice or post-choice frame. All models were fitted to individual time-series extracted from vmPFC and dACC ROI, so as to compute group-level exceedance probabilities, following a BMS procedure similar to that used for behavioral data analysis.

## Acknowledgements

We are grateful to the PRISME platform for help in behavioral and eye-tracking data collection and to the CENIR staff for assistance in MRI data acquisition. The study was funded by a Starting Grant for the European Research Council (ERC-BioMotiv). This work also benefited from the program 'Investissements d'avenir' (ANR-10-IAIHU-06). AL-P received a PhD fellowship from the Direction Générale de l'Armement and from the LabEx Bio-Psy. The funders had no role in study design, data collection and analysis, decision to publish or preparation of the manuscript. We wish to thank Antonio Rangel for suggestions on data analysis and Lionel Rigoux for guidance on computational modeling.

## Additional information

### Funding

| Funder | Grant reference number | Author |
|---|---|---|
| Direction Générale de l'Armement | | Alizée Lopez-Persem |
| LabEx BioPsy | | Alizée Lopez-Persem |
| European Research Council | ERC-BioMotiv | Mathias Pessiglione |
| Agence Nationale de la Recherche | ANR-10-9AIHU-06 | Mathias Pessiglione |

The funders had no role in study design, data collection and interpretation, or the decision to submit the work for publication.

### Author contributions

AL-P, Conception and design, Acquisition of data, Analysis and interpretation of data, Drafting or revising the article; PD, MP, Conception and design, Analysis and interpretation of data, Drafting or revising the article

### Ethics

Human subjects: The study was approved by the Pitie-Salpetriere Hospital ethics committee (protocole C12-69). All subjects were recruited via e-mail within an academic database and gave informed consent to participate and consent to publish before participation in the study.

## Additional files

### Supplementary files

• Supplementary file 1. Table S1 – Activation list for decision value coding in the pre-choice and post-choice frames (GLM 3). Regions are listed that survived voxel-based thresholding of p<0.001 uncorrected, and whole-brain 991 cluster-level FWE correction. [x, y, z] coordinates refer to the Montreal Neurological Institute (MNI) space.

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
