## [Decision Letter]

Thank you for submitting your article "Frames and biases in the human brain: the impact of prior preferences on decision making" for consideration by *eLife*. Your article has been reviewed by two peer reviewers, and the evaluation has been overseen by Reviewing Editor Michael Frank and Timothy Behrens as the Senior Editor. The following individuals involved in review of your submission have agreed to reveal their identity: Laurence Hunt (Reviewer #1); Sebastian Gluth (Reviewer #2).

The reviewers have discussed the reviews with one another and the Reviewing Editor has drafted this decision to help you prepare a revised submission.

Summary:

This manuscript examines the effects of framing choices in terms of a preferred ('default') category of items vs. a less preferred ('alternative') category of items upon choice behaviour (characterised using a drift diffusion model), eye gaze and BOLD fMRI data (with a particular focus on responses in ventromedial prefrontal cortex).

They first show that subjects show a bias towards selecting items from the default category (over and above their preferences expressed before the experiment), both in terms of choices and reaction times. They explain this as a change in starting point of the drift diffusion model. They then use eye-tracking to show that this bias is not explained directly as a consequence of the subjects attending more towards the default item, but instead as a combination of attentional effects on drift rate and a shift in the starting point. Perhaps most strikingly, they show that the default value is present in vmPFC baseline activity, prior to the trial beginning. The key idea is to test whether activity which is often found to reflect chosen minus unchosen value is better thought of as reflecting the value of the default option. They find evidence that this is the case in VMPFC, but not dACC. Overall, this paper addresses a very timely research question, the representation of (choice-related) valuation signals in vmPFC.

Essential revisions:

Both reviewers and the Reviewing Editor were enthusiastic about the contribution, noting that it is a timely approach using a wide array of approaches to address a controversial question, but also noted some essential revisions. These points are amalgamated below.

1) Where does the default bias come from? One possibility is that rather than coming from the decision framing, it comes from a history of choices in favour of that category (of which there will presumably be more towards the default). For instance, if the authors included (chose default(t-1), chose default(t-2), chose default(t-3) etc.) in the logistic regression model, do they see any significant prediction of subjects' choices of the default option on trial t, which decays over trials? Does this reduce the default bias in the constant term?

2) This is also a concern when considering the baseline effect of DV(CAT) in Figure 4. Is the data at this timepoint being influenced at all by the value of the chosen item, or the value of the default item, on the preceding trial?

3) In Results, the authors conclude from their analyses that "fixation duration […] was not influenced by prior preferences". However, I am not fully convinced by their analysis. They report that people look longer at the default option if it is chosen (with a difference of 81 ms), but they look longer at the other option if this is chosen (with a difference of only 41ms). I think the critical question is whether this 81 vs. 41 ms difference is significant or not. In other words, one could set up an ANOVA with factors "Default vs. Non-Default" and "Chosen vs. Unchosen". Apparently, the "Chosen vs. Unchosen" main effect in such an ANOVA would be significant. Now the question is whether the "Default vs. Non-Default" main effect would also be significant or not. Regarding the same topic: It is mentioned that the graphs in Figure 3 are not significantly different from each other. I was wondering what statistical test the authors used here.

4) Related to point #2, the authors are trying to rule out the Krajbich/Rangel aDDM ideas applying here, in that the default biases cannot be explained just by greater value accumulation for the attended option. My concern is that a simpler model could potentially explain much of the gaze data, whereby there could be a direct effect of gaze on choice – where more looking is predictive of higher probability of choice (regardless of direction of causality) – here, gaze time is additive to the value effects on drift rate, rather than multiplicative as in aDDM. (See Cavanagh et al., 2014 JEP-G for an example of where this additive model provided a better fit than the aDDM; Krajbich/Rangel never tested this simpler model). My worry here is that if subjects attend more to the default option than the non-default option, that this could directly drive drift toward the default. That wouldn't show up as well when fitting the aDDM because it assumes that attention modulates the value, but behavioral data show that there is a bias in choice independent of value, so the authors still need the added starting point bias independent of gaze. But if the subjects just look at the default more, and looking directly predicts choice independent of value, then they might find that the default bias is entirely accounted for by more looking… It would be better if the authors added this direct effect as another model in their model comparison (and perhaps a model with this direct effect plus starting point bias). Still, even if this is the case, I'm not sure it diminishes the contribution in terms of what it tells us about vmPFC or default biases (especially since it would be the default preference that would drive the bias in looking, not the other way around).

5) The authors are keen to show that the RT biases arise directly as a consequence of the choice bias. They provide some evidence of this in Figure 2. It would be nice to explore this further. One further way of addressing this would be to ask whether the RT bias goes to 0 at the true point of subjective equivalence between the two options (i.e. where P(default) = 0.5, rather than where |VIT(def) – VIT(alt)| =0). Another way of showing it would be to plot Figure 2 without absoluting the X-axis (i.e. plot positive and negative values for VIT(def)-VIT(alt), as in Figure 2).

6) A concern when examining the effects in Figure 3 is that any propensity to initially saccade towards the default option might be masked. This could produce a 'starting point' like effect in the aDDM, if one assumes attentional deployment is not random (unlike in the aDDM original formulation). An easy way to show that there was no bias in initial saccade direction would be to show the same plot time-locked to stimulus onset, rather than response.

7) Unsurprisingly, the authors cite Boorman et al., J Neurosci 2013 quite extensively, as this also addressed the question of default vs. alternative coding (but didn't have the nice and extensive behavioural modelling of the present paper). However, the conclusion from Figure 4 seems quite different from that paper. In this paper, dorsal anterior cingulate cortex cares about the chosen vs. unchosen value frame (Figure 4—figure supplement 1 bottom), whilst VMPFC cares about the default vs. alternative frame (Figure 4). This seemed to me like the opposite conclusion from Boorman's paper (cf. his Figure 4)…? It would be good for the authors to mention/discuss this.

8) Both reviewers agreed that there was no any information or value that is added to this paper by the decoding analysis. When we already know (from the "standard" analyses) that the vmPFC encodes the value of the default option, it is trivial (almost circular) that we can then also decode preferences from it (it's basically just a question of data quality). Unless the authors can come up with a very convincing argument why this analysis is necessary, they should take it out. Simply doing such an analysis because "we can", and because it is a "fancy" method, is not enough motivation.

---

## [Author Response]

[…]

*Essential revisions:*

*Both reviewers and the Reviewing Editor were enthusiastic about the contribution, noting that it is a timely approach using a wide array of approaches to address a controversial question, but also noted some essential revisions. These points are amalgamated below.*

*1) Where does the default bias come from? One possibility is that rather than coming from the decision framing, it comes from a history of choices in favour of that category (of which there will presumably be more towards the default). For instance, if the authors included (chose default(t-1), chose default(t-2), chose default(t-3) etc.) in the logistic regression model, do they see any significant prediction of subjects' choices of the default option on trial t, which decays over trials? Does this reduce the default bias in the constant term?*

We thank the reviewers for raising this issue, which is indeed important for the interpretation of the default bias. What we showed in the manuscript is that on top of option values, the constant term was significantly positive in the logistic regression used to fit the probability of choosing the default. This constant might capture a prior in favor of the default option, which can be approximated by DV_CAT_ (difference in value between categories), or by the past choices which, on average, should favor the default option. To disambiguate this alternative, we included both DV_CAT_ and past choices (coded 1 for default and -1 for non-default chosen) in the logistic regression. The model was therefore:

logit(Pdef) = Cte + V_IT_(def) + V_IT_(alt) + DV_CAT_ + Choice(t-1) + Choice (t-2) etc.

We first noted that when including DV_CAT_, the constant term was considerably reduced and only bordering significance (from β=0.68±0.13, t(23)=5.40, p=2.10^-5^ to β=0.31 ± 0.16, t(23)=1.94, p=0.06). The DV_CAT_ regressor itself was highly significant (β=0.021±0.006, t(23)=3.53, p=2.10^-3^), suggesting that it was the main factor explaining the default bias. On

the contrary, none of the past choices had a significant effect (all p>0.09, with trends in the opposite direction). In Figure 5 are illustrated regression estimates obtained with the logistic model that only included the last trial (t-1). Similar results were obtained when including more past trials in the GLM. It therefore appeared that the default bias was not generated by the history of choices.

Author response image 1.**DOI:**
http://dx.doi.org/10.7554/eLife.20317.009

This analysis is now mentioned in the Results section of the manuscript, as follows:

“Crucially, the constant was significantly positive (β=0.68±0.13, t(23)=5.40, p=2.10^-5^), bringing evidence for a bias toward the default option. […] Consistently, the constant estimate was not different when restricting the logistic regression to the first choice in a block (β_first_=0.58 ± 0.49, β_all_=0.68±0.13, difference: t(23)=0.97 p=0.44), confirming that the default bias was not resulting from the history of past choices.”

*2) This is also a concern when considering the baseline effect of DV(CAT) in Figure 4. Is the data at this timepoint being influenced at all by the value of the chosen item, or the value of the default item, on the preceding trial?*

We understand the reviewer’s concern: the problem is that DV_CAT_ might be correlated with the default or chosen option value of the preceding trial. We ran another version of GLM2 with the parametric modulators (V_IT_(def_chosen), V_IT_(alt_unchosen), V_IT_(def_unchosen), V_IT_(alt_chosen)) replaced by the same values but for the options presented in the previous trial (the first trial of each block was discarded). In order to correct for DV_CAT_, we also included DV_CAT_ as a parametric modulator before the four item values. We did not find any significant effect of previous trial option values in baseline activity (all p>0.1 for the 4 values at the 2 time points t=-2 and t=0 sec from stimulus onset). Thus, it seems that the values of the options presented in the preceding trial (default or alternative, chosen or not) do not significantly influence vmPFC baseline activity, beyond the shared variance with DV_CAT_.

We have also mentioned this analysis in the Results section of the manuscript, as follows:

“As we realized that the evoked response might have contaminated baseline vmPFC activity at the next trial, we estimated another version of GLM2, with the four parametric modulators – V_IT_(def_chosen), V_IT_(alt_unchosen), V_IT_(def_unchosen), V_IT_(alt_chosen) – replaced by the same values but for the options presented in the previous trial (the first trial of each block was discarded). None of these parametric modulators had a significant effect at the time points preceding option display (t=-2 and t=0 s from stimulus onset). Thus, the values of the options presented in the previous trial did not affect baseline vmPFC activity, beyond the variance that they shared with DV_CAT_.”

*3) In Results, the authors conclude from their analyses that "fixation duration […] was not influenced by prior preferences". However, I am not fully convinced by their analysis. They report that people look longer at the default option if it is chosen (with a difference of 81 ms), but they look longer at the other option if this is chosen (with a difference of only 41ms). I think the critical question is whether this 81 vs. 41 ms difference is significant or not. In other words, one could set up an ANOVA with factors "Default vs. Non-Default" and "Chosen vs. Unchosen". Apparently, the "Chosen vs. Unchosen" main effect in such an ANOVA would be significant. Now the question is whether the "Default vs. Non-Default" main effect would also be significant or not. Regarding the same topic: It is mentioned that the graphs in Figure 3 are not significantly different from each other. I was wondering what statistical test the authors used here.*

We have run the ANOVA suggested by the reviewer, which confirmed our interpretation: there was a significant effect of the ‘Chosen vs. Unchosen’ factor, but no significant effect of the ‘Default vs. Alternative’ factor. We note that the reviewer is concerned with the interaction, which was indeed significant (see below).

The reasons why we did not comment on the interaction is because 1) the two factors are not orthogonal (the default option is more frequently chosen) and 2) it is confounded by the effect of option values on response time (hence total fixation time).

In order to further investigate this interaction and to take into account the effect of DV_IT_, we plotted the proportion of fixation for the default option as a function of DV_IT_, separately for the two types of choices (Black: default chosen; Red: default unchosen). Figure 6 shows that when controlling for decision value, the two situations are symmetrical: the advantage of the chosen option is similar whether that chosen option is the default or the alternative. Thus, the interaction might result from the fact that decision value is on average larger when the default is chosen, yielding a bigger advantage for fixation of the chosen option in that case. Note that the proportions of fixations are symmetrical around the point of subjective equivalence (E) between the two options computed from choices.

Author response image 2.**DOI:**
http://dx.doi.org/10.7554/eLife.20317.010

We have not added this figure in the revised manuscript as we think this interaction on gaze fixations is rather anecdotal relative to the main conclusions of our study. However, we have incorporated the ANOVA in the Results section, as follows:

‘To test the link between prior preference and gaze fixation pattern, we compared the duration of fixation for the default and alternative options, separately for trials in which the default was chosen and unchosen. […] The interaction was significant (F(1,88)=17.45, p=1.10^-4^), reflecting the fact that the default option was more frequently chosen, and with larger decision values.”

*4) Related to point #2, the authors are trying to rule out the Krajbich/Rangel aDDM ideas applying here, in that the default biases cannot be explained just by greater value accumulation for the attended option. My concern is that a simpler model could potentially explain much of the gaze data, whereby there could be a direct effect of gaze on choice – where more looking is predictive of higher probability of choice (regardless of direction of causality) – here, gaze time is additive to the value effects on drift rate, rather than multiplicative as in aDDM. (See Cavanagh et al., 2014 JEP-G for an example of where this additive model provided a better fit than the aDDM; Krajbich/Rangel never tested this simpler model). My worry here is that if subjects attend more to the default option than the non-default option, that this could directly drive drift toward the default. That wouldn't show up as well when fitting the aDDM because it assumes that attention modulates the value, but behavioral data show that there is a bias in choice independent of value, so the authors still need the added starting point bias independent of gaze. But if the subjects just look at the default more, and looking directly predicts choice independent of value, then they might find that the default bias is entirely accounted for by more looking… It would be better if the authors added this direct effect as another model in their model comparison (and perhaps a model with this direct effect plus starting point bias). Still, even if this is the case, I'm not sure it diminishes the contribution in terms of what it tells us about vmPFC or default biases (especially since it would be the default preference that would drive the bias in looking, not the other way around).*

We have fitted the model suggested by the reviewers, with an additive and not multiplicative effect of gaze on the drift rate. As can be seen in Figure 7, the additive model could not account for the default bias (left panel), such that we still needed the shift in the starting point (right panel).

Author response image 3.**DOI:**
http://dx.doi.org/10.7554/eLife.20317.011

In fact, the additive model was doomed to fail since it was not the case that participants look at the default more. We have nonetheless mentioned this additional analysis in the Results section:

“Although using the fixation pattern (with θ) improved the fit, only the prior preference (with S) could explain the decision bias toward the default option. We reached similar conclusions when the advantage for the attended option was additively included in the drift rate, on top of decision value (as in Cavanagh et al., 2014). In fact, gaze fixation pattern failed to produce the default bias simply because the default option was no more looked at than the alternative option.”

*5) The authors are keen to show that the RT biases arise directly as a consequence of the choice bias. They provide some evidence of this in Figure 2. It would be nice to explore this further. One further way of addressing this would be to ask whether the RT bias goes to 0 at the true point of subjective equivalence between the two options (i.e. where P(default) = 0.5, rather than where |VIT(def) – VIT(alt)| =0). Another way of showing it would be to plot Figure 2 without absoluting the X-axis (i.e. plot positive and negative values for VIT(def)-VIT(alt), as in Figure 2).*

We think there is a confusion here. We do not claim that the RT bias arises directly as a consequence of the choice bias. We claim that choice and RT biases have the same origin – the prior preference between categories, which is quantified by DV_CAT_. The prediction that follows on this claim is that choice and RT biases should both go to 0 when DV_CAT_ goes to 0. This is verified in the correlation across blocks (note that DV_CAT_ varies across blocks) shown in Figure 2. It is also verified in the model comparison showing that taking DV_CAT_ as a starting point in the DDM produces both the choice and RT biases.

This does not imply that the RT bias should go to 0 at the point of subjective equivalence, i.e. when the decision value DV_IT_ (not DV_CAT_) is null. We have checked that the DDM does not make this prediction, using simulations (Figure 8). In simulation (1) we fixed the starting point at 0 and we vary the drift rate from -0.1 to 0.1. In this case, corresponding to DV_CAT_=0, there was no bias in choices or in RT. In simulation (2) the starting point was fixed to 0.1 and the drift rate varied as in (1). In this case, we did see a bias in both choice and RT. From simulation (2), we computed the ‘true point of subjective equivalence’ (E=-0.013) for which Pdef=0.5. Then we ran simulation (3), with a starting point still fixed at 0.1 and the drift varying from -0.1-E to 0.1-E. We found that the choice bias was cancelled but that the RT bias survived, including at the equivalence point (when drift was equal to –E).

Author response image 4.**DOI:**
http://dx.doi.org/10.7554/eLife.20317.012

Thus, it should not be expected that taking equivalent subjective values at the item level would cancel the RT bias induced by the preference at the category level. Consistently, we observed in our data that when item values were corrected so as to eliminate the choice bias, the RT bias was still significant (t(23)=8.43, p=4.10^-4^).

We have not included these analyses in the revised manuscript as there was no reason to predict a cancellation or the RT bias with DV_IT_=0, and because it seems rather distractive from the main claim which is about the effect of prior preference (DV_CAT_).

We nonetheless inserted a comment on the fact that choice and RT biases tended to vanish when DV_CAT_ approached zero (Results section:

“We verified this conclusion by testing the correlation across blocks between the posterior means of the 12-free-S model and the prior preference DV_CAT_ (Figure 2). The correlation was significant at the group level (r=0.35 ± 0.07, t(23)=5.12, p=3.10^-5^), strengthening the idea that prior preference was imposing a shift in the starting point that resulted in both choice and RT biases. Thus, the correlation observed between choice and RT biases was driven by variations in DV_CAT_ across blocks, the two biases tending to vanish when DV_CAT_ approached zero.”

*6) A concern when examining the effects in Figure 3 is that any propensity to initially saccade towards the default option might be masked. This could produce a 'starting point' like effect in the aDDM, if one assumes attentional deployment is not random (unlike in the aDDM original formulation). An easy way to show that there was no bias in initial saccade direction would be to show the same plot time-locked to stimulus onset, rather than response.*

When investigating the time course of gaze fixations locked to stimulus onset, we found a strong bias in the initial saccade toward the left option but not toward the default option (see left graph in Figure 9, in which stars indicate time points surviving correction for multiple comparisons obtained through permutation tests). Actually, the initial fixation pattern was essentially spatial: subjects looked first left and then right. The preferential looking of the default option emerged later, corresponding to the fact that it was more frequently chosen. This result is confirmed in the right panel showing that the proportion of fixations between 0 and 200 ms is significantly different from 50% for the left option (t(22)=13.3, p=6.10^-12^) but not for the default option (t(22)=1.28, p=0.21).

Author response image 5.**DOI:**
http://dx.doi.org/10.7554/eLife.20317.013

We have included this observation in the revised manuscript (Results section), as follows:

“Thus, fixation duration was indeed predictive of choice, but was not influenced by prior preference. […] This result was further confirmed by a model comparison showing that fixation duration for each option was better explained (xp=0.999) by a GLM including the unsigned decision value and the choice (chosen vs. unchosen option) than by GLMs including an additional regressor that indicated the prior preference (default vs. alternative option).”

*7) Unsurprisingly, the authors cite Boorman et al., J Neurosci 2013 quite extensively, as this also addressed the question of default vs. alternative coding (but didn't have the nice and extensive behavioural modelling of the present paper). However, the conclusion from Figure 4 seems quite different from that paper. In this paper, dorsal anterior cingulate cortex cares about the chosen vs. unchosen value frame (Figure 4—figure supplement 1 bottom), whilst VMPFC cares about the default vs. alternative frame (Figure 4). This seemed to me like the opposite conclusion from Boorman's paper (cf. his Figure 4)? It would be good for the authors to mention/discuss this.*

The reviewer is absolutely correct: our results stand in contradiction to those reported in Boorman et al., J Neurosci 2013. The conclusions of this paper are indeed that decision values are signaled in the choice frame by the vmPFC, and in the default frame by the dACC. The reason why we did not discuss the contradiction is that we had no good explanation to provide. When taking a closer look at that paper, we realized that the result regarding dACC was not so clear-cut (see their Figure 4). The result regarding vmPFC clearly suggested a post- choice framing, but the activation peak arose much later than what we observed (10 sec vs. 4- 6 sec after option display). It could be that subjects in their design did not set up a default as we define it, which implies anticipating a default response, associated with shortening of response times (RT are not reported in their paper). This would explain why value coding appeared long after the choice was made, and why it was expressed in a post-choice frame only. We now mention the discrepancy and this difference of timing in the Discussion:

“The analysis of the evoked response showed that the vmPFC and ventral striatum encode the decision value in a frame that opposes the default to the alternative option. […] Accordingly, the representation of chosen option value was largely delayed in comparison to our results (peaking 10s after option display), possibly related to the necessity of storing expected values in a learning context.”

*8) Both reviewers agreed that there was no any information or value that is added to this paper by the decoding analysis. When we already know (from the "standard" analyses) that the vmPFC encodes the value of the default option, it is trivial (almost circular) that we can then also decode preferences from it (it's basically just a question of data quality). Unless the authors can come up with a very convincing argument why this analysis is necessary, they should take it out. Simply doing such an analysis because "we can", and because it is a "fancy" method, is not enough motivation.*

The motivation was not that ‘we can’ or ‘it is ‘fancy’, but simply pedagogic. In our experience when presenting this study, Figure 5 helps a lot to understand the link between prior preference and comparison frame. We nonetheless agree that it provides no additional evidence, so we removed the parts regarding decoding analysis from the Methods, Results, Discussion and figure/legend sections.